# Surrounded by Kindred: *Spermophilus major* Hybridization with Other *Spermophilus* Species in Space and Time

**DOI:** 10.3390/biology12060880

**Published:** 2023-06-17

**Authors:** Andrey Tukhbatullin, Oleg Ermakov, Svetlana Kapustina, Vladimir Starikov, Valentina Tambovtseva, Sergey Titov, Oleg Brandler

**Affiliations:** 1Koltzov Institute of Developmental Biology, Russian Academy of Sciences, Vavilova Str. 26, Moscow 119334, Russia; tukhbatullinandrej@gmail.com (A.T.); svetkapust@gmail.com (S.K.); tambovceva@idbras.ru (V.T.); 2Faculty of Physics, Mathematics and Natural Sciences, Belinsky Institute of Teacher Education, Penza State University, Lermontov Str. 37, Penza 440026, Russia; oaermakov@list.ru (O.E.); svtitov@yandex.ru (S.T.); 3Department of Biology and Biotechnology, Institute of Natural and Technical Sciences, Surgut State University, Lenin Avenue 1, Surgut 628412, Russia; vp_starikov@mail.ru

**Keywords:** hybridization, *Spermophilus*, russet ground squirrel, introgression, mitochondrial genome, nuclear genome, speciation, secondary contact zone

## Abstract

**Simple Summary:**

The role of hybridization in biological evolution and cases of hybridogenic speciation is one of the most complicated but also interesting and actively studied topics. Russet ground squirrel *Spermophilus major* is known as a fine biological species according to a number of criteria—morphological, ecological, and behavioral, respectively. Nevertheless, a large body of evidence has accumulated on the hybridization of *S. major* with four neighboring *Spermophilus* species. Our goal was to identify their contribution to the nuclear and mitochondrial genome of *S. major*, and to propose a hypothesis describing the hybridization history in space and time. We found that 36% of *S. major* individuals had extraneous alleles, and every contacting species contributed to *S. major*’s genetic variability. Our data suggested at least five independent hybridization events that were associated with shifts in the species ranges due to paleoclimate changes. Two of them were potentially accompanied by mitochondrial captures with replacement of the mitochondrial genome of one of the hybridizing species. We highlight the potential threat to the existence of *S. major* as a species under the current conditions of population decline along with the simultaneous influx of extraneous genes. It is also particularly important to emphasize the necessity of protecting this inhabitant of unique steppe communities.

**Abstract:**

Among the numerous described cases of hybridization in mammals, the most intriguing are (a) cases of introgressive hybridization deeply affecting the evolutionary history of species, and (b) models involving not a pair of species but a multi-species complex. Therefore, the hybridization history of the russet ground squirrel *Spermophilus major*, whose range has repeatedly changed due to climatic fluctuations and now borders the ranges of four related species, is of great interest. The main aims of this study were to determine the direction and intensity of gene introgression, the spatial depth of the infiltration of extraneous genes into the *S. major* range, and to refine the hypothesis of the hybridogenic replacement of mitochondrial genomes in the studied group. Using phylogenetic analysis of the variability of mitochondrial (CR, *cytb*) and nuclear (*SmcY*, *BGN*, *PRKCI*, *c-myc*, *i6p53*) markers, we determined the contribution of neighboring species to the *S. major* genome. We showed that 36% of *S. major* individuals had extraneous alleles. All peripheral species that were in contact with *S. major* contributed towards its genetic variability. We also proposed a hypothesis for the sequence and localization of serial hybridization events. Our assessment of the *S. major* genome implications of introgression highlights the importance of implementing conservation measures to protect this species.

## 1. Introduction

Numerous studies examining gene flow between species in various groups of animals have shown that hybridization in secondary contact zones is a more frequent event then the occasional crossing of reproductive barriers [1,2,3,4,5,6,7,8]. Hybridization plays one of the key roles in speciation, along with natural selection and genetic drift [9,10,11,12,13,14]. Genome-wide investigations have demonstrated that hybridization has made a big contribution to the formation of modern species [2,15], and traces of remote hybridization persist for millions of years [16,17,18]. Gene flow mainly occurs between closely related species, although today there are some data available on hybridization at the level of genera [19,20]. Most of the analyzable hybridization events of divergent lineages have been associated with relatively young secondary contact zones (which are either modern or formed during Quaternary environmental changes) [21,22,23].

Hybridization in mammals is no longer considered a rare phenomenon [24,25,26,27]. Meanwhile, its influence on the evolutionary history on each particular species is not the same. The traces of hybridization are often only found in contact zones [28,29]. Despite the fact that the sizes and spatial organization of such zones can be different [30,31], their main characteristic is the particular localization of gene flow between species. In such cases, hybridization does not significantly affect the evolution of the contacting species.

In the case of introgressive hybridization, on the contrary, the haplotypes of the donor species are integrated into the gene pool of the recipient species [32], and traces of hybridization can be found far beyond the hybrid zone [33]. In some instances, intense introgression can lead to a complete replacement of the recipient alleles with donor alleles [34], and with further independent evolution, contribute towards the formation of new hybrid species [35,36].

However, in addition to the spatial aspect of hybridization, there is also a temporal aspect. With repeated occurrences of secondary contacts between closely related species, multiple separated in time cases of hybridization may occur as a result. Moreover, each hybridization event can contribute to the genomes of the contacting species [37,38,39]. The logical assumption is that newly acquired alleles that expand the phenotypic variability of the recipient species, especially when they persist in its genome for many thousands of years, should be supported by selection. However, due to the co-evolution of genes, the genome of the recipient species often exhibits a certain resistance. Therefore, cases where an extraneous allele is more evolutionarily advantageous should be extremely rare. In contrast, neutral alleles are much more likely to persist in a “hybrid” genome. They are weakly affected by selection, and their introgression occurs under the influence of demographic factors unique for each species [40]. Evolutionary aspects of hybridization, including the factors and mechanisms of gene introgression, the sequence of multiple hybridizations, and the contribution of each event of inter-specific admixture to genetic variability attract the close attention of researchers, as well as the direction of the introgressive substitution of a part or complete mitochondrial genome of a recipient species. However, these issues still remain insufficiently studied.

Palearctic ground squirrels of the genus *Spermophilus* Cuvier, 1825 are a relatively young group of ground squirrels. A few species of this taxa were formed under allopatry and are poorly differentiated [41]. Fourteen species of *Spermophilus* are colonial hibernating inhabitants of the steppe and meadow ecosystems of open landscapes in most parts of Eurasia [42]. Seven *Spermophilus* species hybridize with each other in different combinations [43,44]. Such a widespread admixture of species makes this genus a promising and convenient model for studying the evolutionary aspects of inter-specific hybridization in mammals. The main object of our research was the russet ground squirrel *Spermophilus major* Pallas, 1778, a species that interbreeds with the largest number of other *Spermophilus* species (including *S. suslicus*, *S. fulvus*, *S. pygmaeus*, and *S. brevicauda*, respectively) in nature [43,44]. Nevertheless, it is a species that is well differentiated by the morphological, bioacoustic, and ecological characteristics from other ground squirrels [41,45,46]. The question of the effect that such an intense admixture has on the *S. major* genome is of undoubted interest for understanding the ways of expanding intra-specific variability, and the mechanisms of maintaining the species identity under the influence of hybridization.

The russet ground squirrel has an extended range [45,47,48] partially divided by the Ural Mountains into the western and eastern parts (Figure 1). In the western part of its range (in the Volga region), the russet ground squirrel forms currently active zones of hybridization with yellow (*S. fulvus*) and little (*S. pygmaeus*) ground squirrels on the left bank of the Volga River [49], and speckled ground squirrels (*S. suslicus*) on the right bank [50]. Meanwhile, the formation of the hybrid zone of *S. major* × *S. suslicus* is determined by the time of the transition of the russet ground squirrel to the right bank of the Volga River, which hardly exceeds a hundred years [51]. Traces of hybridization of *S. major* × *S. suslicus* have only been found in the contact zone on the right bank of the Volga River [50], indicating the presence of the barrier function of the river for the spread of the extraneous haplotypes. Considering the short time of existence, narrow localization, and isolation of this hybrid zone, we assumed that crosses with *S. suslicus* cannot significantly affect the genome of *S. major* in the main range (to the east of the Volga River). Therefore, we excluded the consideration of the hybridization of this pair of species from our study. Population genetic analysis revealed a significant introgression of the mitochondrial haplotypes of *S. fulvus* and *S. pygmaeus* in the Volga populations of *S. major*, located not only in the sympatry zones, but also at a considerable distance from them, both on the right and on the left bank of the Volga River [43,49,52]. More than 50% of the russet ground squirrels living at a distance of about 100 km to the left and right of the Volga River had the mtDNA of one of these species. At the same time, introgression of the haplotypes of nuclear genes, both autosomal and linked to the Y chromosome, was mainly found in the contact zones and was substantially less common [43,53]. Active hybridization of the russet ground squirrel with yellow and little ground squirrels was only located in the Volga region, while sporadic cases of gene introgression [43] were only found further east in the vast sympatric zone on the southern border of the distribution of the russet ground squirrel (Figure 1).

In the eastern part of the *S. major* range, its hybridization with ground squirrels of the *heptneri* form [57] was found (in the interfluve of the Tobol and Ishim, left tributaries of the Irtysh). Ground squirrels with hybrid morphological features, aberrant (=hybrid) acoustic signals, and a hybrid genotype (up to 20% according to RAPD-PCR) were found in this area [58,59]. It should be noted that the systematic position of the ground squirrels of the *Colobotis* group, bordering on *S. major* in the east, is disputable. For a long time, all forms of red-cheeked ground squirrels living in Western Kazakhstan and Siberia were considered as sub-species of the red-cheeked ground squirrel *S. erythrogenys* [44,60]. At the same time, there was a point of view regarding dividing them into *S. erythrogenys* and the Brandt’s ground squirrel *S. brevicauda* [42,47,61], which was supported with molecular data [62]. The *heptneri* form was assigned as a sub-species to *S. major* [41] or to *S. erythrogenys* [42]. Allozyme and molecular data have shown that the *heptneri* form and *S. brevicauda* are phylogenetically close [62]. In this paper, we accepted the interpretation of Helgen et al. [42] on the independence of the species *S. brevicauda* and *S. erythrogenys* and considered *heptneri* as a sub-species of the Brandt’s ground squirrel *S. b. heptneri*. Accordingly, in the east, *S. major* formed a wide zone of hybridization with *S. brevicauda*.

Thus, we observed a wide introgression of extraneous mitochondrial haplotypes in the west and active hybridization in the east. This became the basis for the question of the existence of *S. major*’s own specific mitochondrial genome. Based on the sequencing of cytochrome b (*cytb*) and a fragment of the first subunit of the cytochrome oxidase (*COI*) gene of mtDNA, a wide introgression of the mt genome of *S. brevicauda* into the genome of *S. major* leading to the replacement of the mt genome in *S. major* was suggested [62,63]. A phylogeographic study of *S. major* within the entire range made it possible to determine the species-specific mt genome of *S. major*, which was differentiated to varying degrees from the genomes of other *Spermophilus* species [55]. However, the question of its origin remained open. At the same time, the effect of hybridization on the nuclear genome of *S. major* not only in the contact zones, but also within the entire distribution area of the species, has hardly been studied. Active research was conducted mainly in the sympatric zones or at the boundaries of the range. It can be assumed that the events of genetic exchange involving *S. major* and other *Spermophilus* species in contact with it could occur repeatedly. This was supported by the wide geography of the hybrid zones and the alleged recent rapid expansion of the russet ground squirrel to the west from the Trans-Ural refugium [55].

In our study, we confirmed the occurrence of introgressive hybridization between *S. major* and four neighboring species that were identified previously with the phenotypic and mtDNA data using genotyping with a set of nuclear and mitochondrial molecular markers with materials from all parts of the species’ range. The main goals of our work were to determine the direction and intensity of gene introgression, the spatial depth of the infiltration of extraneous haplotypes into the range of *S. major*, and to clarify the hypothesis of the hybridogenic substitution of mitochondrial genomes in the study group. We present a hypothesis of a sequence of spatiotemporal hybridization events that left a trace on the genomes of the studied *Spermophilus* species.

## 2. Materials and Methods

### 2.1. Samples

The material included tissue samples (claw phalanges, liver, kidneys, and muscles) obtained from ground squirrels deposited in the large-scale research facility “Collection of Wildlife tissues for genetic research” of the Core Centrum of the Koltzov Institute of Developmental Biology, Russian Academy of Sciences (CWT IDB RAS), state registration number 3579666, and the Collection of Penza State University (PSU). Most of the material was collected by the authors. A total of 243 phenotypic *S. major*, 51 *S. brevicauda*, 14 *S. fulvus*, 4 *S. erythrogenys*, and 23 *S. pygmaeus* samples were analyzed, respectively. Specimens defined by phenotype as hybrids of *S. major* × *S. pygmaeus* (1 specimen, locality 30) and *S. major* × *S. brevicauda* (4 specimens, locales 65–68) were included in *S. major*. The characteristics and localities of the materials used are shown in Figure 1 and in the Appendix A. The *S. major* mtDNA control region (CR) haplotype sequences (MW149931-MW150003) [55] and *cytb* sequences of *S. b. iliensis* (AF157863), *S. b. intermedius,* and *S. b. brevicauda* (MH518111, MH518074 to MH518107, and MH518108 to MH518110, respectively) that were deposited in the GenBank database were also used.

Animal work was performed in accordance with the established international protocols as recommended in the Guidelines for Humane Endpoints for Animals Used in Biomedical Research. All experimental procedures were approved by the Ethics Committee for Animal Research of the Koltzov Institute of Developmental Biology RAS in accordance with the Regulations for Laboratory Practice in the Russian Federation; the most recently accepted protocol was numbered 37-25.06.2020. Every effort was made to treat animals as humanely as possible and to minimize their suffering.

### 2.2. DNA Extraction, Amplification, Sequencing, and Restriction Analyzes

Genomic DNA was isolated by either saline [64] or phenol-chloroform [65] deproteinization. PCR was performed with 15 μL of Screen Mix reaction mixture (Evrogen, Moscow, Russia) in a Veriti Termo Cycler platelet amplifier (Applied Biosystems, Waltham, MA, USA). The primers and amplification reaction conditions are listed in the Appendix A.

CR and *cytb* mtDNA were used for species differentiation and the analysis of mitochondrial introgression. To assess the effect of hybridization on the *Spermophilus* nuclear genome, we assessed the genetic markers of the fragments of the male-specific histocompatibility antigen *SmcY*, containing intron 8 and partial cds [53], and biglycan (*BGN*) [66], localized on the Y and X chromosomes, respectively. Sequences of autosomal genes were also assessed: intron 13 of the breakpoint cluster region (*i13BCR*) [25], mechano growth factor (*MGF*), beta-acid glucosidase (*GBA*) [66], protein kinase C, iota (*PRKCI*) [67], thyrotropin (*THY*) [68], proto-oncogene c-myc (*c-myc*) [69], and the ID repeat in intron 6 of transformation-related protein 53 (*i6p53*) [52]. For this purpose, the nucleotide sequences of all the previously mentioned markers, except for *i6p53*, were sequenced for 20 randomly selected samples of *S. major* from the entire range and from 2 (for *S. erythrogenys*) to 43 samples of the other species. Individual samples of *S. brevicauda*, *S. erythrogenys*, and *S. pygmaeus* were selectively sequenced to determine the *i6p53* ID repeat structure. The *i6p53* nucleotide sequences of *S. major* and *S. fulvus* were obtained from the publication of Titov et al. [52]. The most essential criterion for marker selection was the formation of species-specific haplotypes based on the obtained sequences, with no signs of incomplete lineage sorting between *S. major* and other species. For further analysis, we selected five markers that reliably differentiate the russet ground squirrel from most of the neighboring species, as well as have the least haplotypic diversity in the *S. major* sample: *SmcY*, *BGN*, *PRKCI*, *c-myc*, and the *i6p53* ID repeat, which differs in sequence length in most of the species that were evaluated in this work [52]. The genetic markers chosen for analysis yielded a single PCR product. The exception was *PRKCI*, which gave an additional short product of ~200 bp in the samples of little ground squirrel. The analyzed *c-myc* gene sequence included 192 bp of 2 introns and 683 bp of 3 exons, respectively.

Automated sequencing was performed using the ABI PRISM^®^ BigDye™ Terminator v. 3.1 Kit (Applied Biosystems, Waltham, MA, USA), the NovaDye Terminator Cycle Sequencing Kit 3.1 (GeneQuest, Moscow, Russia), the AB 3500 Genetic Analyzer (Applied Biosystems, Waltham, MA, USA), and the Nanofor-05 Genetic Analyzer (Syntol, Moscow, Russia) at the Core Centrum of the Koltzov Institute of Developmental Biology, Russian Academy of Sciences.

Chromatograms were reviewed and manually edited using the Lasergene 11 SeqMan package (DNASTAR, Madison, WI, USA). Due to the presence of a short PCR side product, the end sections of the PRKCI sequence were read from the reverse primer. DNA sequences were aligned using the MUSCLE algorithm [70] in MEGA X [71]. The obtained sequences of diploid markers, in which two overlapping peaks were reproducibly recorded on chromatograms, were encoded and considered as heterozygous. Polymorphic loci were phased based on alleles and combinations of alleles found in the homozygote. The resulting matrices were aligned by trimming the dangling ends.

After alignment in the *BGN*, *PRKCI*, and *c-myc* sequences, species-specific substitutions were determined for which restriction enzymes were selected. These restriction enzymes were selected to produce fragments that distinguished *S. major* from the species that were in contact with it (Figure 2). Species-specific substitutions were identified to search for restriction sites using *BGN*, *PRKCI*, and *c-myc* sequences of at least 10 random *S. major* specimens from different parts of the range, as well as several specimens of *S. brevicauda*, *S. erythrogenys*, *S. fulvus*, and *S. pygmaeus,* respectively. The PCR products of these fragments were then incubated with the appropriate restriction enzymes according to the companies’ protocols. The entire sample of *S. major* was analyzed by restriction analysis for each of the selected markers. Samples of *S. major* showing the presence of a non-specific allele were sequenced. PCR of the *i6p53* ID repeat region was also performed for all samples from the analyzed ground squirrel sample. The *BGN*, *PRKCI*, and *c-myc* restrictase treatments and the *i6p53* ID-repeat PCR products were analyzed using electrophoresis in a 1.5% agarose gel with the GeneRuler 50+ bp Ladder DNA length marker (Eurogen, Moscow, Russia) in the presence of the intercalating dye iridium bromide. Visual detection in the ultraviolet spectrum was performed using the Bio-Rad ChemiDoc MP System gel-documenting system (Bio-Rad, Hercules, CA, USA). The *SmcY* sequence was obtained in all males in a total sample of all the analyzed ground squirrel species.

All newly sequenced nucleotide sequences of the haplotypes were deposited in GenBank with access numbers CR: OR088768-OR088803, *cytb*: OR088804 - OR088840, *SmcY*: OR088841 - OR088846, *BGN*: OR088847 - OR088859, *c-myc*: OQ929560 - OQ929573, *PRKCI*: OQ954111 - OQ954121.

### 2.3. Molecular Data Analyzes

Phylogenetic analysis based on the haplotype variability of the mitochondrial markers was performed for the entire set of samples studied. Selection of the optimal nucleotide substitution model for CR and *cytb* was performed with the ModelTest package in the MEGA X software [71] using the Akaike Criterion (AIC). Stability of the phylogenetic tree nodes was assessed using bootstrap analysis on 1000 replicates. The CR dendrograms were constructed based on the maximum likelihood (ML) method using the HKY+G nucleotide evolution model in the MEGA X software. Identical haplotypes on the tree were compressed to a single branch. The numbering of the CR haplotypes was based on the one previously adopted in the work of Brandler et al. [55]. Based on the clustering of samples on the CR tree for individual *S. major* specimens, a phylogenetic analysis (ML) of *cytb* variation was performed using the Tamura-Nei+G nucleotide evolution model in the MEGA X software. Sequences of geographically distant sub-species of *S. b brevicauda* and *S. b. intermedius* were added while constructing the phylogeny of *cytb*. All trees were rooted using *S. pygmaeus* as the external group, being the most distant taxon. Genetic differences were assessed by pairwise distances (*p*-distance, d_p_).

CR genetic variability was analyzed using Arlequin v. 3.5.2.2 [72], with calculation of the number of haplotypes (*H*), total number of polymorphic loci (*S*), total number of mutations (*η*), haplotype (*h*), and nucleotide (*π*) diversity, along with the average number of pairwise differences (*k*). Spatial expansion tests were also performed using Fu’s (*Fs*) and Tajima’s (*D*) models. Frequency analysis of species-specific alleles of the *SmcY*, *BGN*, *PRKCI*, and *c-myc* genes of the entire *S. major* sample was performed to assess the introgression of extraneous nDNA. Since the aims of the study did not include the assessment of intra-specific variability for the studied markers, most of the samples were analyzed based on restriction analysis of the studied fragments without determining their nucleotide composition.

In addition to the frequency analysis of the *S. major* haplotypes, haplotypic networks were constructed for the *BGN*, *PRKCI*, and *c-myc* genes using the HaplowebMaker software (https://eeg-ebe.github.io/HaplowebMaker/ (accessed on 15 July 2022)) [73]. Haplotype networks were constructed using all the samples that were sequenced after testing with restriction analysis. Thus, the sample for network construction included heterozygous individuals with the addition of one individual carrying each detected unique haplotype in the homozygote, if any. The network was constructed using the median linkage algorithm with an epsilon factor equal to 0. According to J. Doyle [74] “allele pools at each locus are a representation of the overall gene pool, which in turn can be inferred from multilocus genotypes; individuals sharing a common gene pool belong to the same multilocus field for recombination (ml-FFR)”. To date, this approach has been successfully applied to different vertebrate groups [75,76,77,78] to analyze the phylogenetic relationships and heterozygosity of populations.

## 3. Results

### 3.1. Variability of Mitochondrial Molecular Markers

We sequenced one-hundred and fourteen complete nucleotide sequences (1005–1008 bp) of the mtDNA control region: thirty-six—*S. major*, thirty-nine—*S. brevicauda* (thirty-eight—*S. b. heptneri* and one—*S. b. iliensis*), thirteen—*S. fulvus*, three—*S. erythrogenys*, and twenty-three—*S. pygmaeus,* respectively. The complete nucleotide sequences (1140 bp) of *cytb* were also determined for fifty-eight specimens of *S. major*: fifteen of *S. brevicauda* (eight of *S. b. heptneri* and seven of *S. b. iliensis*), five of *S. fulvus*, one of *S. erythrogenys*, and four of *S. pygmaeus,* respectively.

Seventy-two CR haplotypes, including newly sequenced and deposited in the GenBank database were found in the sample of 241 *S. major* specimens. Sixty-six of these haplotypes formed a separate clade on the ML tree (Figure 3a, clade *“major”*). In addition, 10 haplotypes of 29 individuals of *S. b. heptneri* were included in the same clade. Four *S. b. heptneri* specimens had the *S. major* haplotypes *h10*, *h18*, and *h21*. Haplotype *hb05* was found in three specimens of *S. b. heptneri* from population 6b, which was clustered separately from *S. major*, and basally to the haplotypes of *S. b. iliensis*. Three haplotypes found in phenotypic *S. major* (thirteen specimens) clustered with the species-specific haplotypes of *S. fulvus* and three haplotypes (six specimens) clustered with *S. pygmaeus* (Figure 3a). Twenty-five *cytb* haplotypes were found in fifty-eight specimens of *S. major*. Two haplotypes found in phenotypic specimens of *S. major* (ten samples) clustered with the species-specific haplotypes of *S. fulvus* and two haplotypes (five samples) clustered with *S. pygmaeus,* respectively. The topology of the phylogenetic tree (ML) constructed on the basis of 100 *cytb* sequences of five *Spermophilus* species which we newly obtained and took from the GenBank mainly corresponded to the branching CR tree, but haplotype *c25* found in *S. b. heptneri* individuals with *hb05* CR were included in the *“major”* clade (Figure 3b).

*Spermophilus major* specimens with *S. pygmaeus* mtDNA haplotypes were only identified in the Volga populations located upstream of the Volga River from the earlier described hybridization zone [49]. The *S. fulvus* mtDNA haplotypes were detected both in the south of the *S. major* range in localities situated close to the sympatric zone with *S. fulvus* and in the north of the Cis-Ural part of the range. The depth of the infiltration of the yellow ground squirrel mtDNA haplotypes into the range of the russet ground squirrel was found in some cases to exceed the previously described 100 km strip along the Volga River [43,49]. The sample of one population of phenotypic *S. major* (population 69, *n* = 6) included individuals with only the *S. fulvus* haplotypes (Figure 4).

An analysis of the population-genetic variability of CR was performed separately for the entire sample of phenotypic *S. major*, including specimens with species-specific and extraneous haplotypes (A), and for a sample of individuals with only species-specific haplotypes for *S. major* (B) (Figure 3a, *“major”*). The results of the analysis are shown in Table 1. The level of haplotypic diversity (*h*) in the samples was not determined to be statistically different, but the level of nucleotide variability (*k*) and nucleotide diversity (*π*) of sample A was found to be two times greater than in sample B. The values of the tests for spatial expansion were found to be negative in the analysis of both samples, but for sample A the *Fs*-test was deemed to be unreliable, and for sample B the Tajima’s test was determined to be unreliable.

A comparative analysis of genetic distances (Table 2) revealed a significant differentiation of the native *S. major* haplotypes and most of their contacting species by both *cytb* and CR. However, the genetic distances between *S. major*, *S. fulvus*, *S. b. iliensis*, and *S. b. intermedius* were within the dp~0.03 (3%), which is below the 5% empirical potential species level and above the 2% level of intra-specific variation suggested by Baker and Bradley [79]. The *p*-distances of *cytb* between the *S. major* and most *S. b. heptneri* specimens were at the level of the intra-specific variability of *S. major*. The mean pairwise genetic distances between the *cytb* haplotypes of *S. major* in the sample without extraneous haplotypes (clade *“major”*, Figure 3) was d_p_ = 0.002. Three *S. b. heptneri* individuals from population 6b (*hb05*), clustered by CR separately from *S. major* and basally to *S. b. iliensis* (Figure 3), were determined to be genetically distant by *cytb* from *S. major* and *S. b. iliensis* at d_p_ = 0.006 and d_p_ = 0.010, respectively.

### 3.2. Variability of Nuclear Molecular Markers

Nine nuclear markers were assessed to detect gene introgression into the *S. major* genome. Four of them were excluded from the analysis for the following reasons: *THY*—yields a multiple PCR product; *GBA*—not variable in this group of species; and *MGF*—polymorphic in all studied species, carrying several deletions in different haplotypes, which thereby makes it difficult to determine the allelic composition of heterozygous individuals. For *i13BCR* in *S. major*, one widely distributed haplotype shared with *S. fulvus* was found in addition to species-specific haplotypes, which can be explained by both hybridization and synapomorphism. Five markers (*SmcY*, *BGN*, *PRKCI*, *c-myc*, and *i6p53*) were selected for further analysis (Appendix A). Each of the selected markers had species-specific haplotypes and had no sequences showing signs of an incomplete lineage sorting between *S. major* and other species examined in this work.

#### 3.2.1. *SmcY* Variability

We sequenced *SmcY* nucleotide sequences (618 bp) from eighty-six *S. major* males, thirteen *S. brevicauda* males (twelve *S. b. heptneri* and one *S. b. iliensis*), five *S. fulvus,* and six *S. pygmaeus*, respectively (Appendix A). Two *SmcY* haplotypes were found in the *S. major* sample. One of them (*m2*) was unique, found in two individuals, and differs from the haplotypes of all neighboring species. This marker differentiates both detected haplotypes of *S. major* from *S. erythrogenys* by three substitutions, *S. major* from *S. fulvus* by five substitutions, and *S. major* from *S. pygmaeus* by six substitutions, respectively, which corresponds to the previously described [53] (Figure 2a). However, it turned out to be identical in *S. major* and *S. brevicauda*. All individuals analyzed for this marker carried only species-specific haplotypes. The specimen from locality 30 phenotypically determined as a hybrid of *S. major* × *S. pygmaeus* had a *S. pygmaeus*-specific haplotype.

#### 3.2.2. *BGN* Variability

We detected 13 *BGN* haplotypes (Figure 2b) in the analysis of 49 sequences we obtained (751–754 bp). Of these, eighteen were *S. major*, thirteen were *S. brevicauda*, five were *S. fulvus*, two were *S. erythrogenys*, and eleven were *S. pygmaeus*, respectively. The only haplotype (*m*) detected in *S. major* differs from the haplotypes of other species by the substitution of A/G in position 289 and T/C in position 505, respectively. The last substitution formed the recognition site of Bso31 I restrictase in all haplotypes except for haplotype *m*. Restrictase analysis showed that all individuals of *S. major* had the species-specific haplotype *m*. Three of the four *S. major* × *S. brevicauda* hybrids (localities 61–63) had the specific haplotype *brevicauda* (*b1*) in homozygote, while the fourth had haplotype major (*m*) in homozygote. In the hybrid *S. major* × *S. pygmaeus* a haplotype specific to *S. pygmaeus* was found. A haplotype network (“haploweb”) constructed from *BGN* gene sequences (Figure 5a) demonstrated a high polymorphism of this marker in *S. pygmaeus* and *S. brevicauda*. Despite the relatively small sample size of these species, four and three haplotypes were found in them, respectively. All detected heterozygotes for this marker only reflected intra-specific variability.

#### 3.2.3. *PRKCI* Variability

We obtained 97 sequences of *PRKCI* (568–569 bp). Of these, thirty-seven were *S. major*, forty-three were *S. brevicauda* (eight were *S. b. iliensis* and thirty-five were *S. b. heptneri*), six were *S. fulvus*, four were *S. erythrogenys*, and seven were *S. pygmaeus*, respectively. Two *PRKCI* haplotypes were found in *S. major*, differing from the haplotypes of all other species by the A/G transition at the position 343 (Figure 2c). This substitution forms the Ahl I restrictase recognition site in all haplotypes except in *m1*, *m2*, and *b2,* respectively. Haplotype *b2* was only identified in individuals of *S. b. iliensis* from Almaty and differs from the *S. major* haplotypes by the T/A transversion at the position 479. Two species-specific haplotypes were found in *S. fulvus* and *S. brevicauda*. Another haplotype (*k*) was found to be common to the *S. fulvus* and the *S. b. heptneri* forms, which may be a manifestation of ancestral polymorphism. Two species-specific haplotypes were also found in *S. erythrogenys* and one species-specific haplotype in *S. pygmaeus,* respectively.

Restriction analysis revealed the predominance of individuals with only species-specific haplotypes in the entire sample of *S. major* (92.6%). In 13 individuals of phenotypic *S. major* from the Trans-Ural populations (Figure 4), along with haplotypes specific for *S. major,* haplotype *k* was found to be common for *S. fulvus* and *S. brevicauda*. Of the four *S. major* × *S. brevicauda* hybrids determined by the phenotype, one had haplotype *k* common in *S. brevicauda* in homozygote and another had haplotype *S. major* in homozygote. The remaining two hybrids carried the haplotypes of both species in heterozygote. The hybrid *S. major* × *S. pygmaeus* was heterozygous for the species-specific haplotypes. In addition, two individuals from population 29 and one individual each from populations 10 and 39 with the *S. major* haplotype *PRKCI* were found to have an additional PCR product formed by amplification of this marker in *S. pygmaeus*.

The haplotype network based on the sequences of the *PRKCI* gene region (Figure 5b) showed little difference between the haplotypes of all studied *Colobotis* species. Specific haplotypes for each species formed intra-specific unified fields for recombination (FFR). Haplotype *m1* was found to be more common than haplotype *m2* in individuals of *S. major* heterozygous for this marker (Figure 5b), which may reflect the higher frequency of this allele (0.8) in *S. major* populations (Appendix A).

#### 3.2.4. *c-myc* Variability

We detected 13 *c-myc* haplotypes (875 bp) in the analysis of 135 sequences obtained (Figure 2d). Of these, eighty-seven were *S. major*, twenty-five were *S. brevicauda*, ten were *S. fulvus*, two were *S. erythrogenys*, and eleven were *S. pygmaeus,* respectively. One haplotype (*m*) was found in *S. major*, which differed from the haplotypes of the other species by the G/A transition in position 635. This substitution formed the recognition site of restrictase BstDS I in haplotype *m*. Three species-specific haplotypes were found in *S. fulvus*. Another *c-myc* haplotype was found to be common to *S. fulvus* and *S. b. heptneri* (*c*), which may be a manifestation of the ancestral polymorphism as in the case of *PRKCI*. Furthermore, two species-specific haplotypes were found in *S. brevicauda*, and one species-specific haplotype was detected in *S. erythrogenys,* respectively. In the sample of *S. pygmaeus*, four species-specific haplotypes were detected. The *p5* haplotype, which is close to *S. pygmaeus*, was also found in the *S. major* sample, but was not detected in *S. pygmaeus*. We identified this haplotype as belonging to *S. pygmaeus* due to their similarity in nucleotide composition in the conserved region of this sequence.

Restriction analysis showed the predominance of individuals with the species-specific haplotype in the homozygote (72.9%) in the entire sample of *S. major*. In addition, individuals carrying haplotypes specific for *S. pygmaeus* (*p1-5*) and the haplotype common to *S. fulvus* and *S. b. heptneri* (*c*) were found both in heterozygous (12.9% and 8.8%, respectively) and in homozygous (2.1% each) states. The haplotype *erythrogenys* was found in the heterozygote with *major* and *brevicauda* in 2.1% and 1.25% of individuals, respectively. Two of the four hybrids of *S. major* × *S. brevicauda* had haplotype *c* in homozygote, with one of these hybrids having the haplotypes *c* and *b1*, and the other being heterozygous for haplotypes *c* and *e*. The hybrid *S. major* × *S. pygmaeus* had haplotypes of both species.

The haplotype network based on the sequences of the *c-myc* gene region (Figure 5c) showed little difference between the *S. major*, *S. brevicauda*, and *f2* haplotypes of *S. fulvus*. The greatest number of heterozygotes was found between *S. major* and *S. pygmaeus*, most of which (71%) contained the *p5* haplotype found only in the *S. major* populations (Figure 2d).

#### 3.2.5. *i6p53* Variability

Twelve *i6p53* sequences between 100 and 173 bp in length (including primers) were sequenced. Of these, seven were *S. brevicauda*, one was *S. erythrogenys*, and four were *S. pygmaeus,* respectively. The primary structure of the *i6p53* ID repeats differed in all the species studied, except for *S. major* and *S. b. heptneri* (Appendix A). PCR analysis made it possible to differentiate the *S. major* haplotypes from the other species, with the exception of *S. b. heptneri*. All individuals in *S. major* sample, as well as *S. major* × *S. brevicauda* hybrids, had haplotypes common to both *S. major* and *S. b. heptneri*. The hybrid *S. major* × *S. pygmaeus* had haplotypes of both species. No haplotypes typical of *S. major*/*S. b. heptneri* were found in the other species.

#### 3.2.6. Consensus Sequence Analysis

A haplotype network (“haploweb”) based on the combined sequence of *BGN*, *c-myc*, and *PRKCI* genes (Figure 6) showed a significant divergence between *S. major* and the other species. The haplotypes of *S. major* formed a single recombination field, with the vast majority of links going to the species-specific haplotype. The haplotypes of three of the four phenotypic hybrids of *S. major* × *S. brevicauda* formed a single ml-FFR with *S. brevicauda*. The remaining hybrid was found to be a part of the ml-FFR of *S. major*. The hybrid *S. major* × *S. pygmaeus* was also not included in the ml-FFR of *S. major* but was located closer to *S. pygmaeus*.

### 3.3. Analysis of S. major Haplotypic Diversity

Based on the data on the allelic composition of the studied molecular markers, genotyping of all individuals in the total sample of *S. major* was performed. Only species-specific haplotypes of the studied markers were found in 63.7% of the russet ground squirrels. Among the ninety *S. major* individuals with extraneous haplotypes, eleven (12.2%) were hybrid in two or more markers, four of which were phenotypically defined hybrids. More than half of the *S. major* individuals with extraneous haplotypes (52.2%) had *S. fulvus* and/or *S. brevicauda* haplotypes, 43.3% had *S. pygmaeus* haplotypes, and 8.9% had *S. erythrogenys* haplotypes, respectively. In addition, two *S. major* individuals in the west of the range had both *S. fulvus* and *S. pygmaeus* haplotypes, and two individuals in the east had both *S. erythrogenys* and *S. brevicauda* haplotypes.

The *S. fulvus* haplotypes were found in populations 18, 19, and 69 located in the center of the western part of the range, whereas *S. pygmaeus* haplotypes were only found in the Volga region (populations 70–72) (Figure 4a). The mtDNA haplotypes of *S. brevicauda* and *S. erythrogenys* were not found even in the secondary contact zone of *S. major* and *S. brevicauda*. However, haplotypes of all four species were detected in the aggregate set of nDNA markers. The greatest number of extraneous haplotypes was detected for the *c-myc* (16%), half of which were haplotypes specific to *S. pygmaeus*. Among all markers analyzed in *S. major*, 5.6% of haplotypes of the other species were found (Table 3).

## 4. Discussion

The modern genome of *S. major*, both nuclear and mitochondrial contains borrowed haplotypes, thus reflecting the history of its hybridization with neighboring species. Extraneous haplotypes are widespread in many parts of the range of the russet ground squirrel (Figure 4). However, different species contribute differently to the diversity of the *S. major* gene pool. The cumulative contribution of *S. brevicauda* and *S. fulvus* accounts for most of the extraneous haplotypes observed in the *S. major* genome (Table 3). We summarized the contributions of these species because of the ancestral polymorphism found in them for two markers, *PRKCI* and *c-myc*. However, we can hypothesize with a high probability for which of these species was a donor of one or the other haplotype in the populations where they were found. This was determined by their geographic location in relation to the contact zones. Accordingly, we assumed that the *PRKCI* haplotype k in the populations of the southeastern macroslope of the Southern Urals was obtained from *S. fulvus*, and the introgression of this haplotype and the *c-myc* haplotype in the northern and eastern populations occurred as a result of hybridization with *S. brevicauda*. In this case, *S. brevicauda* contributed more to the genetic diversity of *S. major* compared to *S. fulvus* (2.3% vs. 0.75%, respectively). A comparable contribution (2.2%) to the genetic diversity of *S. major* was made by *S. pygmaeus*, despite the fact that it was the most phylogenetically distant among the species hybridizing with *S. major*. Despite the wide distribution of extraneous haplotypes in *S. major* populations, their cumulative contributions to the genetic diversity of the species was only ~6%. However, in *S. major* individuals, *PRKCI* amplification detected an additional PCR product characteristic only of *S. pygmaeus*, suggesting a greater contribution of this and other species to the *S. major* genome. Our data indicate repeated substitutions of the mitochondrial genome in *S. major* and *S. brevicauda*. The clustering of the mitochondrial trees (Figure 3) and the values of the obtained genetic distances (Table 2) point to the fact that these introgression events are limited to the contact of *S. major* with the forms of *S. b. heptneri* and *S. b. iliensis* (or their common ancestor). It is therefore likely that originally, as a result of introgressive hybridization, the mt genome of the ancestral form of *S. b. heptneri*/*iliensis* was replaced by the mt genome of *S. major*. This was evident by the fact that two forms of *S. brevicauda*, *S. b. brevicauda,* and *S. b. intermedius* formed a separate branch on the mt-trees, while *S. b. iliensis* was clustered together with *S. major*. The position of *S. major* on the mt phylogeny as the youngest species of the *Colobotis* group was consistent with the revealed variability of the nuclear markers. The haplotypes of *S. major* by three independent nuclear markers were characterized by the absence of synapomorphies uniting all other *Spermophilus* species and show relatively low variability (Figure 2b-d). However, our data are insufficient to reject the assumption of replacement of the *S. major* mt genome by *S. brevicauda* [63].

We have shown the association of the mitochondrial haplotypes of both the CR and *cytb* of *S. b. heptneri* with the eastern group of the *S. major* haplotypes previously identified by Brandler et al. [55]. This, along with the absence of *heptneri* haplotypes in the western part of the *S. major* range, supports the recent re-substitution of the *S. b. heptneri* mitochondrial genome for the *S. major*’s. The CR haplotype *hb05* found in three individuals of *S. b. heptneri* from Andreevka (population 6b, Figure 4a) is probably ancestral to this sub-species (Figure 3, Appendix A). However, the small sample of specimens carrying this haplotype, as well as the discordance of the clustering of these specimens on CR and *cytb* trees, suggests the necessity to clarify this assumption.

The analysis of the spatial depth of the infiltration of extraneous haplotypes into the *S. major* range requires a comprehensive approach. The shape of the range, localization, and mutual location of contact zones, and the existing physical and geographic barriers should be taken into account. The modern range of the russet ground squirrel is almost completely divided by the Ural Mountains, leaving a relatively narrow passage in the south (Figure 1). As shown earlier, the Ural Mountains are currently an almost impenetrable barrier to gene flow between the western and eastern groups of the *S. major* populations [55]. The eastern and western parts of the range differ both in terms of ecological conditions and in the set of contacting species. Therefore, we considered it expedient to consider separately the Cis-Ural (western) and Trans-Ural (eastern) parts of the *S. major* range.

### 4.1. Hybridization in Space

#### 4.1.1. Cis-Ural Part of the *S. major* Range

Our data indicates the deep infiltration of *S. fulvus* mitochondrial genome elements into the western populations of *S. major* over 200 km from the modern sympatric zone of these species. Previous studies [43] found traces of *S. fulvus* mtDNA introgression into *S. major* genome in a limited band on both sides of the Volga River up to the latitude of Kazan. Combining our results with these data, we can assume the dispersal of russet ground squirrel individuals that received *S. fulvus* mtDNA in the hybrid zone and their descendants to the north along the valley parallel to the riverbed. As was shown earlier, the russet ground squirrel is attracted to the banks of water bodies, and due to this fact, river channels are natural ways of dispersal for this species [51,80].

Further dispersal of distant hybrids eastward could also have occurred along the right tributaries of the Volga–Kama basin. This was supported by the finding of mt haplotypes of *S. fulvus* in the populations of the russet ground squirrel in the north of the Cis-Ural part of the range (populations 18, 19, 69) at a distance of about 200 km from the Volga (Figure 4a). The founder effect resulting from this dispersal strategy was evident in the high proportion of individuals with the *S. fulvus* haplotypes in the *S.major* populations, such as in population 69, where the total sample was composed of such individuals (*n* = 6). In addition, the hybridization of *S. major* × *S. fulvus* apparently occurs not only in the Volga region, but also in other parts of the Cis-Ural overlapping zone of their ranges, as evidenced by the detection of the *S. fulvus* mt haplotype introgression in the south of the *S. major* range in the Poduralsky Plateau (population 29). In contrast to mtDNA, there was no extensive introgression of the *S. fulvus* nDNA haplotypes in the Cis-Ural populations of the russet ground squirrel. This is consistent with the finding of limited introgression of the nuclear haplotypes in only the mixed populations of *S. major* and *S. fulvus* in the Volga region, along with the single case of *SmcY* haplotype introgression of *S. fulvus* in the *S. major* population south of Aktobe [52,53]. The absence of the introgression of nuclear genes may be due to the washout of the yellow ground squirrel nDNA haplotypes during repeated backcrosses in russet ground squirrel populations, which may be additionally accelerated by the selection against extraneous haplotypes.

As in the case of hybridization of the russet and yellow ground squirrels, hybridization of *S. major* × *S. pygmaeus* in the southern Volga Region leads to the introgression of *S. pygmaeus* mtDNA northward along both banks of the Volga River [49]. However, despite the great contribution of the *S. pygmaeus* to the introgression of the mt haplotypes (2/3 of the introgressive haplotypes in the *S. major* genome in the Volga Region [49]), we did not detect the spread of the *S. pygmaeus* mt haplotypes eastward into the *S. major* range. This may be due to the selection against the mt haplotypes of the *S. pygmaeus* in *S. major* × *S. pygmaeus* hybrids due to a significant mito-nuclear discordance [81,82] since they are phylogenetically distant species [63].

In contrast to mtDNA haplotypes, *c-myc* haplotypes of the *S. pygmaeus* were common throughout the Cis-Ural part of the *S. major* range, as well as in the eastern macroslope of the Southern Urals (Figure 4a) within the distribution of the western group of populations. The low frequency of these haplotypes, as well as the absence of mitochondrial markers throughout the range except for the Volga region, indicates a different age of mitochondrial and nuclear introgression events. We hypothesize that the mtDNA introgression of the little ground squirrel in the Volga region is the result of recent hybridization. At the same time, the wide distribution of *S. pygmaeus c-myc* haplotypes in *S. major* populations in the absence of other nuclear markers outside the contact zones may be explained by its possible neutrality. In this case, the observed distribution of *S. pygmaeus c-myc* haplotypes could have formed in accordance with the demographic model of introgression [40]. Conditions conforming to this model might have arisen during the previous spatial expansion of the russet ground squirrel from the East to the West [55].

We detected the introgression of the haplotypes of other *S. pygmaeus* nuclear markers into the genome of the russet ground squirrel in only one phenotypic hybrid of these two species south of the Ural Mountains in the Mugodzhar Hills. The previously described introgression of the *S. pygmaeus* nuclear gene haplotypes did not extend beyond the hybrid zone of these species in the Volga region [53]. The modern infiltration of the *S. pygmaeus* nDNA haplotypes into the *S. major* range may be limited by their decreased frequency and disappearance during repeated backcrossing in the same way as the *S. fulvus* nuclear haplotypes.

#### 4.1.2. Trans-Ural Part of the *S. major* Range

We found no introgression of the mt markers of other species in the eastern part of the *S. major* range. The geographic infiltration of the nuclear haplotype *S. brevicauda c-myc* and the *PRKCI* haplotype k into the *S. major* has been found to reach the Urals. We assumed that *S. brevicauda* was a donor of the *PRKCI* haplotype *k* in the Tobol–Ishim interfluve, since there is a hybridization zone with this species. In the Chelyabinsk region (population 50, Figure 4a, and in Appendix A), the donor of this haplotype was also presumably *S. brevicauda*, but not *S. fulvus*, as *c-myc* haplotypes of *S. brevicauda* have also been found in this population. Probably, the introgression of the *S. fulvus* nuclear haplotypes in the Trans-Urals is limited to the infiltration of the *PRKCI* haplotype k into the *S. major* population of the southeastern macroslope of the Southern Urals.

The *c-myc* haplotypes of *S. erythrogenys* sensu stricto found in populations of *S. major* of the Kurgan region (57, 64, and 69 in Figure 4b, respectively, and Appendix A) indicate the hybridization of these species in the past. Today, the range of *S. b. heptneri* divides the ranges of *S. major* and *S. erythrogenys*. Therefore, it is unlikely that *S. major* received these haplotypes “in transit” through *S. b. heptneri*, as previously described, for example, by triad hybridization in Darwin’s finches [83]. We found no haplotypes of either nuclear or mtDNA of *S. erythrogenys* in samples of *S. b. heptneri*. No traces of *S. pygmaeus* DNA introgression were found in the eastern part of the *S. major* range, which may indicate the absence of hybridization between these species in the Trans-Ural region.

### 4.2. Hybridization over Time

As shown above, hybridization with the neighboring species had a major impact on the russet ground squirrel genome. Hybridization events of *S. major* with different species occurred independently at different times in different parts of the range. Here, we propose a hypothesis for the sequence and localization of these hybridization events.

Paleogeographic events that determined the fragmentation of the ranges of the steppe species and the following secondary contacts of the diverging phyletic lineages had both common features and peculiarities in the Trans-Ural and Cis-Ural territories. The Trans-Ural part of the modern *S. major* range and the neighboring territories of Western Siberia and Northern Kazakhstan were under the influence of the Quaternary cyclic Turgai spillways (defined as periodic discharges of glacial waters of the Irtysh basin into the Aral basin). “The system of now degraded old runoff troughs ... testifies to a threefold large-scale watering of the plains of the Southern Turgai and Northern Aral Sea region” [84]. Around 400,000 years ago during the late Pleistocene period, coniferous forests prevailed in the Pre-Ural part of the range, reaching the Caspian Sea, which were subsequently replaced about 125,000 years ago with the broad-leaved forests [85,86]. Steppe vegetation communities were widely distributed in this area by the end of the Mikulino interglacial around 90,000 years ago [86,87]. Fossils of *S. superciliosus*, which is considered an ancestral form of *S. major* and morphologically almost indistinguishable from it, are known from the Middle Pleistocene [41,88]. The ancestral range of the russet ground squirrel was thereby evidently reduced, fragmented, and restored under the influence of the Pleistocene paleoclimatic changes. The expansion of the arid and semi-arid areas in periglacial zones during cooling was replaced by the formation of extensive watercourses during warming and vice versa. These dynamics favored speciation under conditions of the isolation of ancestral ranges and subsequent secondary contacts between the closely related species of the steppe ecosystems.

Apparently, one of the earliest hybridization events, traces of which we found, was the contact between *S. major* and *S. erythrogenys* in Western Siberia, which left *c-myc* haplotypes of *S. erythrogenys* in the *S. major* genome. This could have occurred if the range of *S. erythrogenys* extended to the left bank of the Irtysh River. However, the paucity of the paleontological records and the poor study of ground squirrels in the territory of the Tobol–Irtysh interfluve do not allow us to determine the paleo-range boundaries of these species. The contact between *S. major* and *S. erythrogenys* may have been interrupted during the middle Pleistocene period as a result of the formation of one of the early Turgai spillway [84].

Probably, the next event of introgressive hybridization in the history of the russet ground squirrel was the contact with *S. brevicauda* sensu lato, which led to the complete replacement of the mtDNA of one of the species. Equal genetic distance between all species of the *Colobotis* group (excluding the more distant *S. erythrogenys*) suggests the simultaneous radiation of *S. fulvus*, two forms of Brandt’s ground squirrel—*S. b. brevicauda*, *S. b. intermedius,* and the *S. major*—*S. b. heptneri/iliensis* group. Hybridization of *S. major* × *S. brevicauda* sensu lato apparently occurred after the division of *S. brevicauda* into the *S. b. brevicauda*, *S. b. intermedius,* and *S. b. heptneri/iliensis* lineages, respectively, as indicated by the different genetic distances between these forms and *S. major*. However, this hybridization event apparently preceded the separation of *S. b. heptneri/iliensis* into two separate forms, which is supported by the proximity of the CR haplotypes of *S. b. iliensis* and the presumably discovered ancestral haplotype of *S. b. heptneri* (Figure 3a). We assumed that the ranges of these species were again separated by the Turgai spillway formed during the Late Karga period [84], after which *S. major* and *S. b. heptneri/iliensis* evolved independently.

Later, *S. major* dispersed westward, integrating the *S. pygmaeus* nDNA haplotypes into its genome. The lack of *S. pygmaeus* mtDNA introgression traces of this time could be explained by the faster sorting of mitochondrial lines compared to nuclear ones, such in the spiral-horned antelopes *Tragelaphus* [89]. It is known that the russet ground squirrel inhabited the Urals [90] and the east of it [88] as early as the Pleistocene. In the paleontological records of the Cis-Ural region, russet ground squirrels were found throughout the middle-late Pleistocene but disappeared in this area by the end of the Pleistocene—beginning of the Holocene [91], probably due to afforestation and/or the swamping of the steppe habitats. Phylogeographic data indicate the re-population of this area by the russet ground squirrel, which apparently occurred as a result of rapid simultaneous expansion bypassing the Ural Mountains from the south [55]. During the last glacial maximum which occurred around 18,000–20,000 years ago, the boreal forest belt stretched from the southern tip of the Urals through the Mugodzhar Hills far to the south, and separated the Trans-Ural and Volga periglacial steppes. Therefore, the westward expansion of *S. major* could not have occurred earlier than the first third of the Holocene which occurred around 6000 years ago, when a continuous latitudinal strip of steppes was formed [85]. The migration routes of *S. major* must have passed through the territories inhabited by *S. pygmaeus*, the first finds of which in the South Urals were dated around 8,500 years ago [90,92]. During the *S. major* invasion into the *S. pygmaeus* range, rare events of their hybridization could have occurred in the Southern Urals, Mugodzhar Hills, or Poduralsky Plateau, with further westward dispersal of the hybrids along with the pure *S. major*. In this case, the localities of *S. pygmaeus* fossils coincide with the region where introgression of the *pygmaeus* haplotypes into the *S. major* genome were found (Figure 4a). A similar situation has been previously described for the genus *Homo* (*H. sapiens* × *H. neanderthalensis* hybridization [93]) and is consistent with the demographic model of introgressive hybridization [40].

The most recent evolutionarily significant event for *S. major* in the Trans-Ural part of its range can be considered as the hybridization of *S. major* × *S. b. heptneri*, which resulted in an almost complete replacement of the *S. b. heptneri* mt genome by the *S. major* ones. This may have occurred in the case of hybridization, when the short-tailed ground squirrel moved into the territory of the russet ground squirrel, resulting in the subsequent displacement of the last [40]. To date, a wide hybrid zone between these two species persists in the Kurgan region, as evidenced by the spatial distribution of these hybrids (Figure 4).

At the same time, a hybrid zone of *S. major* with *S. pygmaeus* and *S. fulvus* was probably formed more recently in the south of the Volga Region [51]. In spite of the fact that the ranges of all these species overlapped throughout the southern border of the *S. major* range from the Volga River to the east, active hybridization was only observed in a small contact zone located in the Saratov Region. This was probably due to the presence of ecological niches in this region that were suitable for all three species, thereby contributing to the formation of joint settlements and hybridization.

## 5. Conclusions

Modern approaches to the study of inter-specific interactions allow us to operate not only with a pair of hybridizing species but also consider entire species complexes. The gene flow and direction of allele introgression may differ in different species complexes. For example, asymmetric mtDNA introgression from several species into one universal mtDNA recipient species in a group has been achieved with a complex of western North American chipmunk species [94]. However, the introgression of nDNA in this group of species appeared to be insignificant. Another example is the species complex of the Tibetan highland pika (genus *Ochotona*) [95], which was formed around a central wide-area species. This species is a universal haplotype donor for more narrow-area species living on the periphery of its range. In this species complex, the introgression of both mitochondrial and nuclear haplotypes was detected.

The *Spermophilus* ground squirrel species that we studied also form a complex of hybridizing species formed around *S. major*. All peripheral species in contact with *S. major* contribute to its genetic variability. This admixture is not limited to the introgressive hybridization of the closely related species of *Colobotis*, which diverged relatively recently as a result of radiation from a common ancestor. The relatively evolutionarily distant little ground squirrel has also made a significant contribution to the general gene pool of the russet ground squirrel. The inclusion of a relatively small number of extraneous mtDNA haplotypes into the gene pool of the russet ground squirrel increases its nucleotide diversity by an order of magnitude (Table 1). In this complex, the central species, *S. major*, played the role of both a recipient and donor species of mtDNA in different periods of time. In the western part of the range, it accumulates mtDNA from two species, *S. fulvus* and *S. pygmaeus*. In the eastern part, it transfers its mtDNA to *S. brevicauda*. As for the introgression of nDNA, *S. major* is mainly a recipient. The available data indicate a low efficiency of reproductive barriers between the russet ground squirrel and its neighbors. However, in this system of past and present hybridizations, despite the active accumulation of extraneous alleles, we do not observe the blurring of the *S. Major* species boundaries. All examined ground squirrel species are deemed as good “Linnaean” species [47,53]. All studied individuals of *S. major* were well differentiated from other species by their morphological features. Most of them, including those with extraneous haplotypes, had a species-specific *S. major* acoustic signal [96], which is a stable species attribute in ground squirrels [97]. The only exceptions were ground squirrels from a narrow zone of hybridization with *S. b. heptneri* [58], and one hybrid *S. major* × *S. pygmaeus* (our unpublished data) having aberrant acoustic signals. The level of genetic mitochondrial differentiation within this group of species could not be a meaningful indicator of species differences due to past mitochondrial captures. The true level of genetic divergence in this group may be revealed based on a whole nuclear-genome phylogenetic study.

The first mitochondrial capture during the late Pleistocene hybridization of *S. major* and *S. brevicauda* apparently led to the formation of an independent haplogroup *iliensis* within *S. brevicauda* as a result of subsequent genetic divergence, as in *Odocoileus* deer [36]. A second recent mitochondrial capture that occurred between these species, combined with the mutual introgression of nuclear genes, initiated the formation of a hybrid genome of the *heptneri* form that continues to the present day. The hybrid nature of the *heptneri* genome seems to be related to the intermediate character of the morphological variability of this form and is consistent with the assumption of Gromov et al. [41] on its hybrid origin. Our hypothesis of the directions of mt genome introgression needs additional research. To clarify the direction of mitochondrial capture, it is therefore necessary to compare the position of these species and their sub-specific forms on mtDNA and nDNA phylogenetic trees, as has been performed for the feline Felidae [1].

In current times, the russet ground squirrel *S. major* is experiencing a deep depression: its abundance is rapidly declining, population sizes are shrinking, and small colonies are disappearing [80]. Under these conditions, an increase in intra-specific genetic diversity due to introgressed extraneous genes may contribute to the conservation and recovery of the species. On the other hand, it can lead to a significant change in the species gene pool as a result of genetic drift with a hypothetically possible catastrophic reduction in the number and habitat area of the species.

## Figures and Tables

**Figure 1 biology-12-00880-f001:**
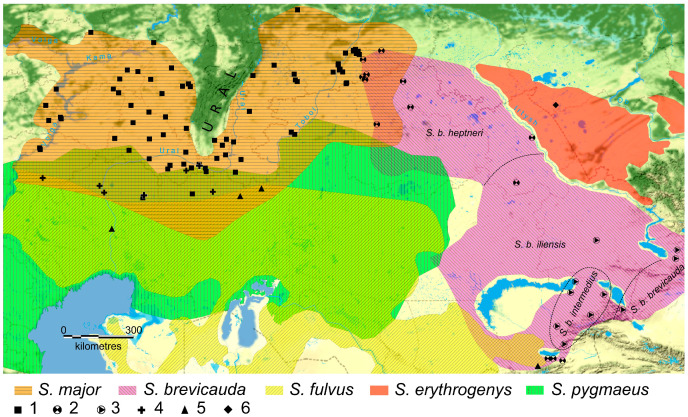
The ranges of the five studied *Spermophilus* species. Color indicates the ranges of the studied species (according to [44,47,54,55] with our modifications). Symbols on the map indicate the locations where material was collected: 1—*S. major*, 2—*S. brevicauda*, 3—specimens of *S. brevicauda* from GenBank, 4—*S. pygmaeus*, 5—*S. fulvus*, and 6—*S. erythrogenys*. The dotted line indicates the conditional boundaries of the *S. brevicauda* sub-species distribution according to Matrosova et al. [56] with additions.

**Figure 2 biology-12-00880-f002:**
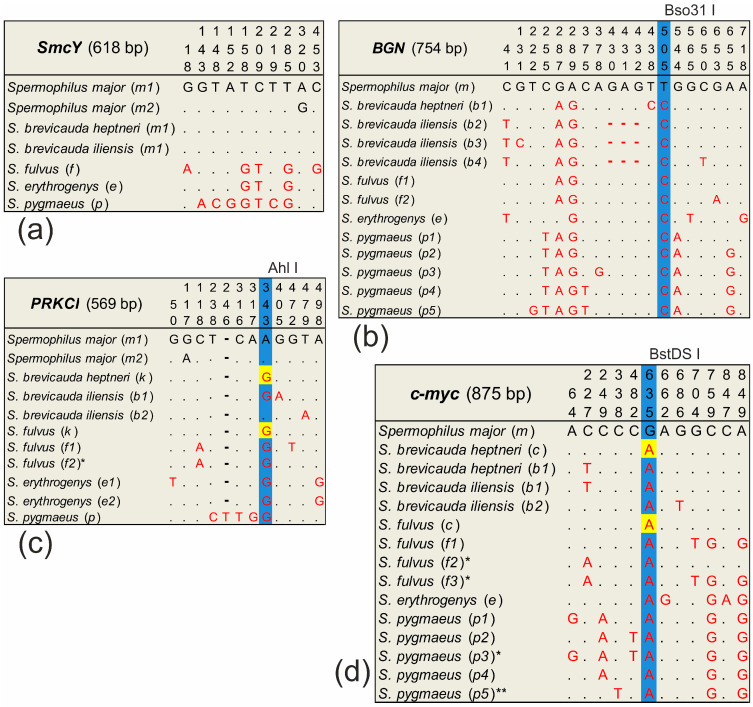
Primary structure of the marker sequences: (**a**) *SmcY*, (**b**) *BGN*, (**c**) *PRKCI*, and (**d**) *c-myc*. Only sites with nucleotide substitutions are marked (complete haplotype sequences are deposited in GenBank). The length of the analyzed fragment is given by the longest sequence without taking into account the cut ends and primers. Haplotypes common to *S. fulvus* and *S. brevicauda* are highlighted in yellow. Restriction sites are marked in blue; the enzyme is indicated above the table. * haplotype only found in the heterozygous state. ** haplotype of *S. pygmaeus* detected in a sample of *S. major* but not detected in *S. pygmaeus* (explanation in the text).

**Figure 3 biology-12-00880-f003:**
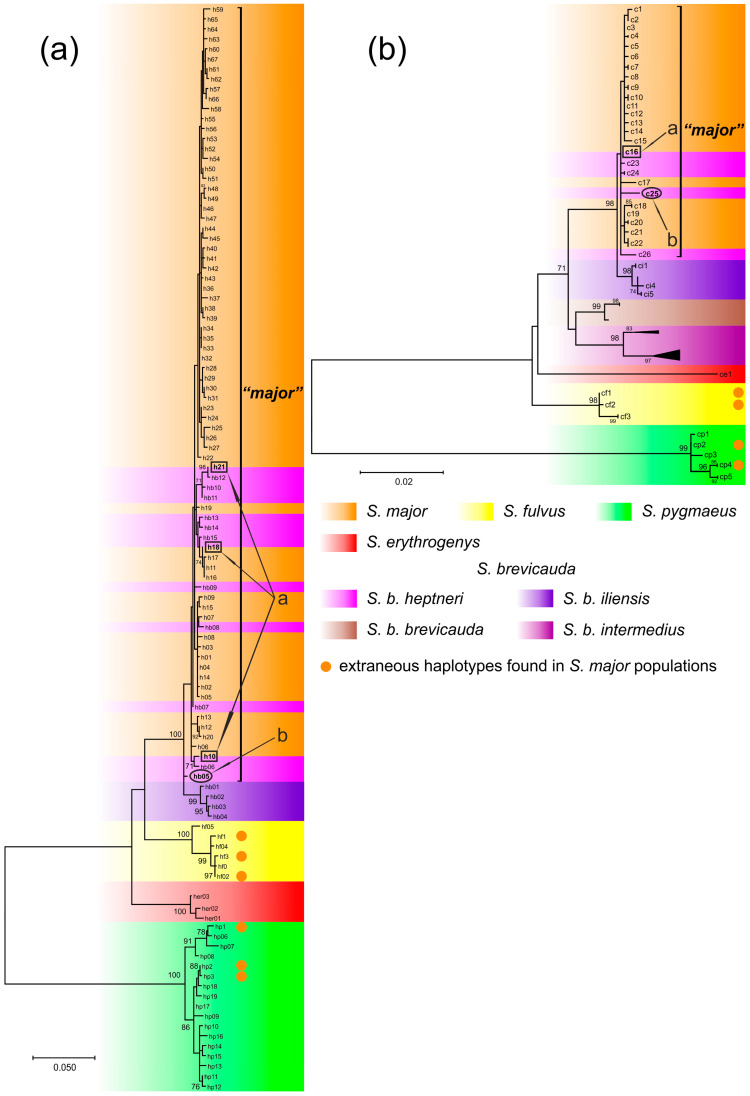
ML-dendrogram of *S. major* and contacting *Spermophilus* species based on *p*-distances: (**a**) CR mtDNA (HKY+G+I model); and (**b**) *cytb* (Tamura-Nei+G model). Phenotypic species and sub-species of *S. brevicauda* are highlighted in color. The arrows point to: a—the haplotypes common to *S. major* and *S. b. heptneri*; and b—the presumed ancestral haplotype of *S. b. heptneri* (explanation in the text). The haplotype names correspond to the Appendix A; unnamed *cytb* haplotypes belong to sequences obtained from GeneBank and are absent in Appendix A. Numbers at the nodes indicate bootstrap support indices; values <70 are not specified.

**Figure 4 biology-12-00880-f004:**
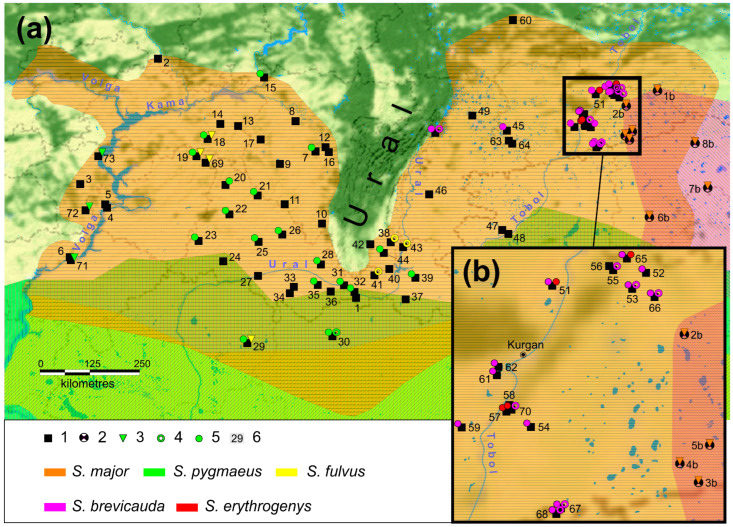
Map of the distribution of the introgressed haplotypes in the range of *S. major*: (**a**) the entire distribution area of *S. major*; (**b**) the contact zone *S. major* and *S. b. heptneri*. The color indicates the species affiliation of the haplotype. The symbols indicate: 1—locations of the capture of *S. major*, the absence of additional symbols indicates the absence of introgressed haplotypes in the population; 2—locations of the capture of *S. b. heptneri*. Locations of the findings of introgressed haplotypes of the studied markers: 3—mtDNA; 4—*PRKCI*; 5—*c-myc*, with the color of the icon reflecting the species specificity of the marker; and 6—population numbers according to Appendix A.

**Figure 5 biology-12-00880-f005:**
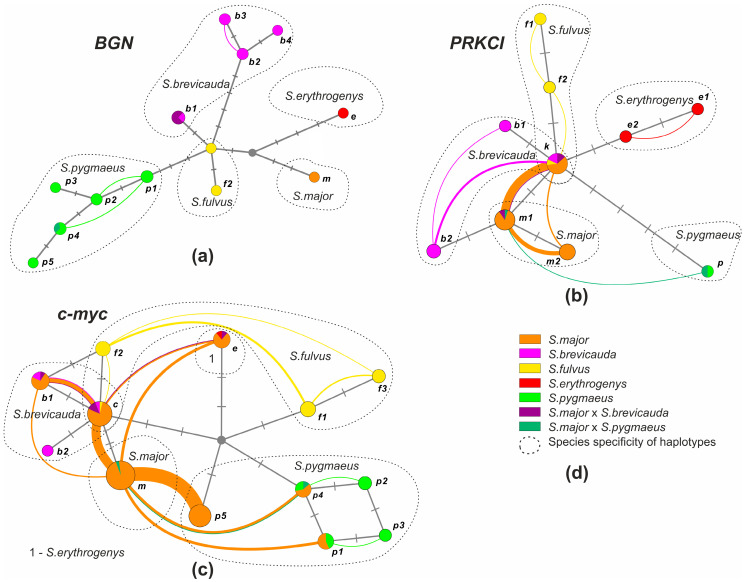
Haplotype networks of nuclear markers generated by HaplowebMaker: (**a**) *BGN*; (**b**) *PRKCI*; (**c**) *c-myc*; and (**d**) symbols. Color indicates species and inter-specific hybrids determined using phenotypic traits. The circles denote haplotypes; the size of the circles is proportional to the distribution of haplotypes in hybrid individuals + one homozygous individual, if any. Haplotype names correspond to Figure 2. The dotted line shows the haplotypes corresponding to each of the species. Nucleotide substitutions are indicated by dashes. The arcs connect the haplotypes found in one individual; the thickness of the arcs is proportional to the number of hybrids with this combination of haplotypes in the sample.

**Figure 6 biology-12-00880-f006:**
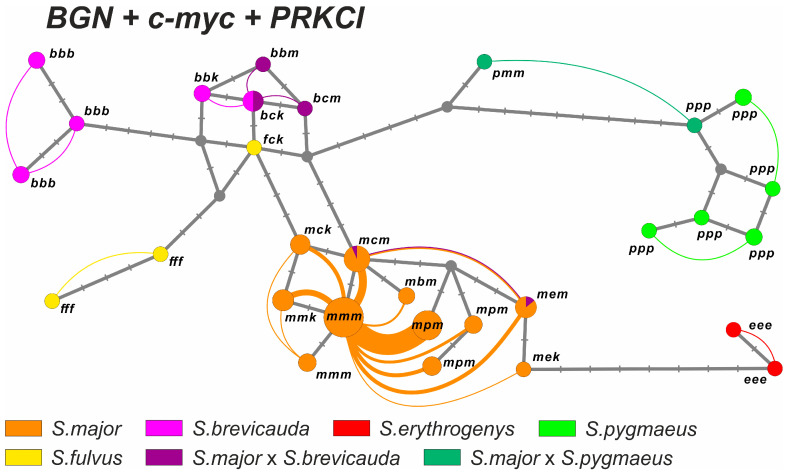
Haplotype network generated by HaplowebMaker for the consensus sequence of *BGN*, *c-myc*, and *PRKCI* gene fragments. The circles denote haplotypes; the size of the circles is proportional to the distribution of haplotypes in the hybrid individuals + one homozygous individual, if any. The phenotypic definition of the taxa is highlighted in different colors and corresponds to Figure 4. Nucleotide substitutions are indicated by dashes. The arcs connect the haplotypes found in one individual; the thickness of the arcs is proportional to the number of hybrids with this combination of haplotypes in the sample. Letters denote the species identity of each marker haplotype: *m*—*S. major*, *b*—*S. brevicauda*, *f*—*S. fulvus*, *e*—*S. erythrogenys*, *p*—*S. pygmaeus*, *c*—haplotype *c-myc*, common for *S. brevicauda* and *S. fulvus*, and *k*—haplotype *PRKCI*, common for *S. brevicauda* and *S. fulvus*.

**Table 1 biology-12-00880-t001:** Genetic diversity of *S. major* populations based on CR variability. Sample **A** contains all CR haplotypes found in phenotypic *S. major*, while sample **B** only contains the haplotypes forming a separate monophyletic clade, *“major”* (Figure 3a). The characteristics of genetic diversity corresponding to the designations in the table are listed in the “Materials and Methods”. SD—standard deviation, and *p*—probability value.

Sample	*n*	*H*	*S/* *ƞ*	*h* ± SD	*π* ± SD	*k*	Tajima’s *D* (*p*-Value)	*Fs* (*p*-Value)
**A**	241	72	204/229	0.979 ± 0.002	0.0179 ± 0.009	18.335	−1.4016 (0.01)	−9.6147 (0.07)
**B**	223	66	80/85	0.976 ± 0.003	0.0078 ± 0.004	7.914	−1.162 (0.14)	−24.43 (<0.01)

**Table 2 biology-12-00880-t002:** Mean pairwise genetic distances (*p*-distances) between *S. major* and contacting species/sub-species according to CR (above the diagonal) and *cytb* (below the diagonal). Sub-species of *S. brevicauda* were considered independently.

No.	Species/Sub-Species	1	2	3	4	5	6	7	8
1	*S. major*		0.012	0.021	–	0.036	0.039	0.049	0.081
2	*S. b. heptneri*	0.003		0.017	–	0.033	0.039	0.046	0.084
3	*S. b. iliensis*	0.007	0.007		–	0.031	0.045	0.050	0.089
4	*S. b. brevicauda*	0.024	0.024	0.026		–	–	–	–
5	*S. b. intermedius*	0.031	0.031	0.034	0.027		0.047	0.056	0.094
6	*S. fulvus*	0.033	0.032	0.035	0.032	0.044		0.049	0.091
7	*S. erythrogenys*	0.054	0.054	0.055	0.050	0.054	0.051		0.088
8	*S. pygmaeus*	0.106	0.106	0.107	0.106	0.109	0.102	0.113	

**Table 3 biology-12-00880-t003:** Percentage of extraneous alleles in the *S. major* gene pool.

Species/Alleles	CR	*SmcY* ^1^	*BGN* ^2^	*c-myc*	*PRKCI*	*i6p53*	Consensus
*S. brevicauda*	0	–	1.29	6.3 ^3^	3.6 ^3^	0	3 ^3^
*S. fulvus*	5.4	0	0			0	
*S. erythrogenys*	0	0	0	1.7	0	0	0.4
*S. pygmaeus*	2.5	1.2	0.26	8	0.2 (1) ^4^	0.2	2.2 (2.4) ^4^
Total	7.9	1.2	1.55	16	3.8 (4.6) ^4^	0.2	5.6 (5.8) ^4^

^1^ Calculated for males only. ^2^ Calculated jointly for males and females, taking into account the number of alleles in each sex. ^3^ Calculated for both species jointly due to ancestral haplotype polymorphism. ^4^ Calculated considering *PRKCI* sequence-independent introgression of an additional PCR product during *PRKCI* sequence amplification in *S. pygmaeus*.

## Data Availability

All newly sequenced nucleotide sequences of the haplotypes were deposited in GenBank NCBI with access numbers OR088768–OR088859; OQ929560–OQ929573; OQ954111–OQ954121.

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
