# Peer review of "Surrounded by Kindred: *Spermophilus major* Hybridization with Other *Spermophilus* Species in Space and Time"

_biology, 2023, doi:10.3390/biology12060880_

Round 1

Reviewer 1 Report

Studying the role of hybridization in speciation and the effect of hybridization on adaptation and speciation are both important in evolutionary biology. This MS is based on a wonderful system, S. major and four neighboring species, to investigate the effect of hybridization on the genome of S. major by determining the direction and extent of gene introgression.

I think that the study system here is nice. However, I wonder whether the number of nuclear loci used here (only five nuclear markers) is enough to fulfill the aim of this MS. If no whole-genome sequencing data were used, I suggest the authors to refine their aim of the study to just test for or confirm the occurrence of introgressive hybridization between S. major and four neighboring species identified previously by the phenotypic data, or something like that.

Overall, the current data analyses are ok, but I prefer to use other methods (e.g. IM model and ABC model) to test for the occurrence of introgression and also the direction of introgression. In addition, these methods (e.g. IM model and ABC model) can help to rule out the possibility of ILS (incomplete lineage sorting of the ancestral polymorphisms) which exhibits similar patterns with introgression. In fact, my main concern is that how authors rule out the possible effect of ILS on those nuclear markers.

Author Response

Studying the role of hybridization in speciation and the effect of hybridization on adaptation and speciation are both important in evolutionary biology. This MS is based on a wonderful system, S. major and four neighboring species, to investigate the effect of hybridization on the genome of S. major by determining the direction and extent of gene introgression.

We are thanking a lot for the interest to our work.

I think that the study system here is nice. However, I wonder whether the number of nuclear loci used here (only five nuclear markers) is enough to fulfill the aim of this MS. If no whole-genome sequencing data were used, I suggest the authors to refine their aim of the study to just test for or confirm the occurrence of introgressive hybridization between S. major and four neighboring species identified previously by the phenotypic data, or something like that.

We agree with this comment. We understand that to achieve the initial aim, a representative increase in the number of molecular markers, and preferably whole genome sequencing data, is needed. We changed the aim of our work as suggested. (Page 4, lines 165-168)

Overall, the current data analyses are ok, but I prefer to use other methods (e.g. IM model and ABC model) to test for the occurrence of introgression and also the direction of introgression. In addition, these methods (e.g. IM model and ABC model) can help to rule out the possibility of ILS (incomplete lineage sorting of the ancestral polymorphisms) which exhibits similar patterns with introgression. In fact, my main concern is that how authors rule out the possible effect of ILS on those nuclear markers.

We used ML analysis to estimate the introgression of mitochondrial haplotypes by their location on dendrograms. We selected short nuclear markers that could be qualitative species markers for genotyping of individuals in our S. major sample. One of the criteria for assessing the suitability of a marker for this purpose was the absence of common haplotypes for S. major and other species, which could be the result of ILS. Therefore, five of the nine tested markers were selected which had no common haplotypes of S. major with the other species. We added clarifications to the relevant parts of the text (Page 5, lines 215-218; Page 12, lines 392-394). Several approaches to model nuclear marker variability have been tested by us. However, small fragment lengths and few substitutions did not allow us to obtain significant results. We are grateful to the reviewer for advice on using alternative models. However, it seems to us that these models (IM model and ABC model) require a larger data set, such as whole-genome data. We will follow this reviewer's advice in future work with extended data that we plan.

Reviewer 2 Report

This is a really nice paper that assesses hybridization across multiple species.  It is a complex situation but the authors have used a rigorous methodology and have successfully described the multiple hybridization events and outcomes.

I have a couple of minor comments for consideration:

1) Maybe use Kimora 2-parameter model for cytb so use can directly compare to other mammal datasets when discussing divergence amounts.  Also K2P could be used in establishing time since divergence values forming events (more on this in number 3).n Add divergence models to figure legends fore ease of reference.

2) Maybe add a concluding paragraph on species status. Some folks may argue this is one species (based on the significant amounts of hybridization across multiple taxa...).  Build a case for multi-species being present.

3) Use time since divergence to estimate age of hybridization events.  You present a very nice phytogeography picture.  Use genetic divergent dates to test!

English is really good.  Just a few things to fix.

Author Response

Answers to the Open Review 2.

This is a really nice paper that assesses hybridization across multiple species.  It is a complex situation but the authors have used a rigorous methodology and have successfully described the multiple hybridization events and outcomes.

We are grateful to the reviewer for the high evaluation of our research.

I have a couple of minor comments for consideration:

1) Maybe use Kimora 2-parameter model for cytb so use can directly compare to other mammal datasets when discussing divergence amounts.  Also K2P could be used in establishing time since divergence values forming events (more on this in number 3).n Add divergence models to figure legends fore ease of reference.

We used p-distances to estimate genetic distances because most phylogenetic and phylogeographic studies of Eurasian ground squirrels use this measure of divergence. This allowed us to compare our data and the data from the literature. We agree that the K2P model proposed by the reviewer is more informative in phylogenetic studies. However, our study was not aimed as a phylogenetic study of the whole group, therefore we limited ourselves to the mentioned approach. We added a divergence model to figure 3 legend as suggested. (Page 9, lines 330-331)

2) Maybe add a concluding paragraph on species status. Some folks may argue this is one species (based on the significant amounts of hybridization across multiple taxa...).  Build a case for multi-species being present.

We agree with this comment. The sentence about the non-burring of the S. major species boundaries was included in the second paragraph of Conclusions. We continued this paragraph with a summary of the arguments for species independence of S. major and the other species studied. (Page 21, lines 790-800; Page 26, lines 1079-1082)

3) Use time since divergence to estimate age of hybridization events.  You present a very nice phytogeography picture.  Use genetic divergent dates to test!

We agree with the reviewer that the use of divergence times from the molecular data could significantly enhance the arguments supporting our phylogeographic hypothesis. However, there are currently no published well-supported molecular phylogenies of the genus Spermophilus with estimates of divergence times for the species/subspecies we studied. An estimate of these divergence times is possible if the phylogeny of the entire genus (or most of the species) was reconstructed and well-dated paleontological fossils are available. We have initiated such research, but it is still far from completion. It is not possible to estimate the divergence times in the nodes of the shortened tree including only the studied species due to the lack of a sufficient number of paleontological fossils. This was partially mentioned in the manuscript (Page 19, lines 705-707). The use of averaged rates of substitution accumulation (molecular clocks) could produce very large errors on the short time intervals within which events of hybridization of our ground squirrels took place, as we believe. On the other hand, estimating divergence times from mitochondrial trees may not be correct because of the mitochondrial genome substitutions we hypothesize. True divergence times may be calculated when phylogeny is constructed from whole-genome sequencing data, which has not been performed yet. Given the above considerations, we decided not to use molecular divergence times in our hypothesis.

Comments on the Quality of English Language

English is really good.  Just a few things to fix.

Thank you for the high evaluation of English quality of our manuscript. We have fixed some shortcomings in the text (Page 2, lines 72-73, 83, 87; Page 4, line 171; Page 16, line 569; Page 17, line 592).
